# Hypertrophic Cardiomyopathy versus Storage Diseases with Myocardial Involvement

**DOI:** 10.3390/ijms241713239

**Published:** 2023-08-26

**Authors:** Anna Burban, Szymon Pucyło, Aleksandra Sikora, Grzegorz Opolski, Marcin Grabowski, Agnieszka Kołodzińska

**Affiliations:** 1First Department of Cardiology, Medical University of Warsaw, ul. Banacha 1A, 02-097 Warszawa, Poland; anna.burban@wum.edu.pl (A.B.); spucylo@gmail.com (S.P.); sikora.alk@gmail.com (A.S.); grzegorz.opolski@wum.edu.pl (G.O.); marcin.grabowski@wum.edu.pl (M.G.); 2Doctoral School, Medical University of Warsaw, 81 Żwirki i Wigury Street, 02-091 Warsaw, Poland

**Keywords:** hypertrophic cardiomyopathy, Fabry disease, Pompe disease, Danon disease, genetics, gene therapies

## Abstract

One of the main causes of heart failure is cardiomyopathies. Among them, the most common is hypertrophic cardiomyopathy (HCM), characterized by thickening of the left ventricular muscle. This article focuses on HCM and other cardiomyopathies with myocardial hypertrophy, including Fabry disease, Pompe disease, and Danon disease. The genetics and pathogenesis of these diseases are described, as well as current and experimental treatment options, such as pharmacological intervention and the potential of gene therapies. Although genetic approaches are promising and have the potential to become the best treatments for these diseases, further research is needed to evaluate their efficacy and safety. This article describes current knowledge and advances in the treatment of the aforementioned cardiomyopathies.

## 1. Introduction

Heart failure (HF) is one of the leading causes of death worldwide. HF is a clinical presentation, a set of symptoms that manifest impaired heart function. Among the cardiac dysfunctions leading to heart failure are cardiomyopathies. They involve structural and functional abnormalities of the heart muscle, in the absence of coronary artery disease, hypertension, valvular disease, and congenital heart disease sufficient to cause the observed myocardial abnormalities. The most common type of cardiomyopathy is hypertrophic cardiomyopathy (HCM). The diagnosis is based on an increase in the thickness of the left ventricular muscle which cannot be explained solely by its abnormal strain. HCM, however, needs to be differentiated from other systemic diseases that may proceed with myocardial involvement.

This article aims to present cardiomyopathies that require differentiation from HCM in the primary diagnosis because they occur with myocardial hypertrophy. This article, in addition to HCM, focuses on storage diseases that result in myocardial involvement and hypertrophy: Fabry disease, Pompy disease, and Danon disease. Importantly, in all these diseases, significant recent advances have been made in gene therapies, which may be the future of treatment for these conditions. However, each disease has different pathogenesis and genetics. They have been compared briefly in Table 1. The currently used and experimental methods, including technological advances in genetics and genomics as well as novel molecular and cellular mechanisms, are presented.

## 2. Hypertrophic Cardiomyopathy (HCM)

HCM is a genetic disorder which is caused by mutations in genes coding for the structure of the sarcomeres of the heart muscle. Abnormal sarcomeres lead to myocardial hypertrophy. The estimated incidence of HCM ranges from 1:200 to 1:500, depending on established criteria [1,2,3]. The disease is characterized by increased thickness of the ventricular wall (≥15 mm), which cannot be explained by the left ventricular afterload state alone, and results in left ventricular hypertrophy (LVH) [4]. It can lead to impaired ventricular diastolic function, left ventricular outflow tract obstruction (LVOTO), imbalance between myocardial oxygen supply and demand, and cardiac arrhythmias [5,6].

The clinical manifestation of patients with HCM has a wide spectrum, from asymptomatic patients to sudden cardiac death (SCD) among young patients and athletes [7,8,9]. In the course of the disease, the myocardium can be ischemic, which explains symptoms such as chest pain, exercise intolerance, and exertional dyspnea [10]. In patients with HCM, heart failure (HF) occurs without the volume overload and pulmonary congestion that are typically associated with HF. Moreover, HCM symptoms are not necessarily related to the severity of LVH or the degree of LVOTO [4]. As a consequence of LVH, patients often have supraventricular and ventricular arrhythmias. Non-sustained ventricular tachycardias (NSVT) are detected in 20–30% of patients diagnosed with HCM [11]. They can lead to ventricular fibrillation and ultimately to SCD, and young adults and athletes are at the biggest risk [8,9]. Implantable cardioverter-defibrillators (ICDs) are effective in preventing SCD in high-risk patients by interrupting potentially fatal ventricular tachyarrhythmias. Their widespread use has led to a significant reduction in HCM mortality (a 10-fold decrease) [12]. In contrast, until now, no pharmacological intervention is known to effectively prevent SCD [11,13].

Apart from implanting ICDs for preventing SCD, pharmacological and surgical treatment is available for patients with HCM. As a result of these developments, both morbidity and mortality have significantly decreased in the last 10–20 years [7]. Non-obstructive HCM patients should have pharmacological therapy (which consists mainly of beta blockers or verapamil) administered at the onset of HF symptoms. Non-obstructive HCM patients should have pharmacological therapy administered at the onset of HF symptoms. When left ventricular ejection fraction (LVEF) ≥ 50%, currently recommended drug classes are beta blockers, verapamil, diltiazem, and low-dose diuretics. On the other hand, when LVEF < 50%, beta-blocker, angiotensin-converting enzyme inhibitor, angiotensin receptor blocker, angiotensin receptor neprilysin inhibitor, sodium–glucose cotransporter 2 inhibitor, and low-dose diuretics, are recommended [4]. It is recommended that HF patients receive pharmacological treatment on an empirical basis in accordance with current guidelines in order to improve their functional capacity, reduce symptoms, and prevent the progression of the disease. In symptomatic patients with LVOTO, the aim is to alleviate symptoms of heart failure. It is recommended that symptomatic LVOTO patients receive beta-blockers as a first-line treatment, and if ineffective or the patient has a contraindication or does not tolerate them, verapamil or dilitiazem can be used. If symptoms persist, disopyramide or mavacamten can be added to beta-blockers, verapamil, or dilitiazem. On the other hand, for symptomatic patients without LVOTO, treatment focuses on controlling arrhythmias, reducing LV filling pressures, and treating angina [4]. For patients with LVOTO, surgical treatment can be offered as well. Transaortic septal myectomy is the preferred treatment option for most patients with limiting HF symptoms refractory to medical therapy and secondary to mechanical LVOTO [14]. Myectomy relieves outflow obstruction. The use of invasive treatment to reduce LVOTO should be considered in patients with a left ventricular outflow track (LVOT) gradient > 50 mm Hg, moderate-to-severe symptoms (New York Heart Association (NYHA) functional class III–IV), and/or recurrent exertional syncope despite maximally tolerated drug therapy [4]. Alcohol septal ablation is the primary alternative to myectomy for severely symptomatic patients who are not optimal operative candidates. However, it should not be performed in children, adolescents, and young adults [14].

### 2.1. Sarcomere Genes

Patients with HCM can be divided into “sarcomere-positive” and “sarcomere-negative”. Those “sarcomere positive” have an identifiable pathogenic or likely pathogenic genetic variant. Pathogenic variants in more than 20 genes are currently known to be associated with HCM [15]. The presence of these pathogenic variants is found in 30% to 60% of HCM patients. A substantial number of patients are classified as “sarcomere-negative” as they do not have any identified genetic etiology of HCM [15]. Clinical heterogeneity and incomplete penetrance of HCM-related variants raise the possibility of nongenetic or environmental factors that may modify the phenotype of HCM. However, they have not been fully characterized yet. The presence of sarcomere mutation can be a powerful predictor of adverse outcomes in patients with HCM. Those patients with pathogenic/probably pathogenic sarcomere variants had a 2-fold higher risk of adverse events compared to patients without mutation [16]. Pathogenic sarcomere variants are associated with younger age at diagnosis, non-sustained ventricular tachycardia, thicker interventricular wall, and less frequent apical HCM [17]. Furthermore, they have a higher degree of myocardial fibrosis, which is associated with an earlier onset of advanced HF and SCD [18,19]. In addition, they are characterized by more severe diastolic dysfunction [19]. In contrast, sarcomere-negative HCM patients follow a much more benign clinical course compared to those with sarcomere-positive HCM [20,21]. 

Sarcomeres are the main contractile unit of the heart. They consist of thick filaments of myosin and thin filaments of actin, closely associated with a complex of regulatory troponin, α-tropomyosin, and cardiac myosin-binding protein C. Up to 40% to 60% of HCM patients carry pathogenic variants in one of the eight core sarcomeric genes, which encode contractile proteins of cardiac sarcomeric myofilaments and Z-band [22]. These pathogenic variants are inherited in an autosomal dominant manner. They affect the following sarcomere proteins: myosin heavy chain (MYH7), myosin C-binding protein (MYBPC3), cardiac troponin T (TNNT2), cardiac troponin I (TNNI3), α-tropomyosin (TPM1), regulatory myosin light chain 2 (MYL2), essential myosin light chain (MYL3), and actin (ACTC1) [23,24]. Among “sarcomeric-positive HCMs,” the majority of pathogenic variants (70–80%) are localized to MYH7 and MYBPC3 [23,25]. To date, more than 1500 different disease-causing variants have been discovered [26]. Certain variants are “high-risk” in that they are associated with a more unfavorable prognosis, such as mutations in the MYH7 converter region [27,28] or the c.2737+1 (IVS26) G>T mutation in the MYBPC3 gene (leading to exon 26 skipping, resulting in severe ventricular hypertrophy and a high risk of SCD) [29]. Patients with pathogenic variants in the MYH7 gene are more likely to have advanced heart failure and a worse prognosis than those with pathogenic variants in the MYBC3 gene [16]. It is also worth noting that pathological variants of thin filament protein genes (TNNT2, TNNI3, ACTC1) are associated with a higher risk of SCD in childhood and advanced LV dysfunction and heart failure in adulthood [18]. Finally, patients carrying two or more different disease variants in sarcomeric genes are at higher risk for fatal arrhythmias and adverse disease progression [30].

Recently, Park et al. conducted a study with a genome-first approach; the total sample size of 41,759 subjects was analyzed by whole exome sequencing (WES), and the results were then compared with their clinical data [31]. Subsequent analysis showed that only 38.5% of patients diagnosed with HCM (with a pathogenic variant in MYBPC3 or MYH7) had genetic testing performed at diagnosis. This means that in more than 60% of these patients, pathogenic variants could be detected by a genome-first approach [31]. It showed also the importance of evaluating novel variants in human genome research, both predicted loss-of-function (pLOF) and predicted deleterious missense (pDM) [31]. pLOF variants in MYBPC3 were strongly associated with the occurrence of HCM, while in the MYH7 gene it was pDM variants that were associated with the disease. In addition, 26.7% of carriers of the pLOF variant in the MYBPC3 gene without a diagnosis of HCM had clear features of myocardial enlargement or hypertrophy in echocardiography [31].

In addition, rare non-sarcomere gene variants (termed pathogenic in HCM) have been identified. These are a small number of genes with established associations with HCM. Some of them are manifesting some unique phenotypic features; CSRP3 missense mutation is clinically presented as HCM and mild skeletal muscle disease [32], and patients with ALPK3 mutation have been characterized by a high prevalence of apical and concentric patterns of LVH and a low prevalence of LVOTO [24]. For some of them, the specific phenotypic pattern is not sure; however, there is strong evidence with HCM (PLN, FLH1) or moderate evidence (FLNC) [33].

### 2.2. Myosin Inhibitors (Mavacamten and Aficamten)

The first drug to be registered specifically for the treatment of patients with HCM (at the clinical trial stage) is mavacamten. Mavacamten (YK-461) is a molecule that acts as an allosteric modulator of cardiac β-myosin. It reduces contractility and improves ventricular compliance by inhibiting actin–myosin cross-bridges in a reversible manner [34,35]. Thus, it suppresses the development of hypertrophy and fibrosis, which is key in the development of HCM [36]. Initially, promising data came from studies in animal models. In a mouse model with a myosin mutation (R403Q), administration of mavacamten was followed by a reduction in ventricular hypertrophy [37]. In a study in a cat model of HCM, mavacamten was shown to alleviate LVOTO [35]. Ongoing clinical trials are still trying to answer the question of whether mavacamten is effective in improving functional capacity in patients with both obstructive and non-obstructive HCM. The data concerning the benefit for patients with HCM are inconsistent. It seems that the use of mavacamten shows higher benefits for patients with obstructive HCM compared to those with non-obstructive HCM.

Mavacamten was tested in patients with non-obstructive HCM during the MAVERICK-HCM randomized clinical trial. In the second phase of the trial, mavacamten therapy was proven safe for patients in this group [38]. However, it has not demonstrated clinical efficacy. Patients with non-obstructive HCM showed no significant improvement in peak oxygen consumption (pVO2) or NYHA class [38]. Nevertheless, after 16 weeks of mavacamten, there was a significant reduction in both N-terminal B-type natriuretic propeptide (NT-proBNP) and cardiac troponin I (cTnI) levels compared to the placebo group [38]. This could indicate a positive effect of the drug on reducing myocardial workload. Nonetheless, more studies are needed to know the role of mavacamten in non-obstructive HCM.

To date, there have been two randomized phase 3 trials involving patients with obstructive HCM: VALOR-HCM and EXPLORER-HCM. Previously, the phase 2 studies showed that mavacamten was generally well tolerated and had a similar safety profile to placebo with mostly mild (80%) adverse events [39]. They also showed that the drug could reduce post-exercise LVOTO, improve exercise capacity, and reduce dyspnea symptoms in patients with obstructive HCM [39]. In the phase 3 EXPLORER-HCM clinical trial, patients with LVOT gradient > 50 mm Hg and NYHA class II-III symptoms received mavacamten or placebo. Compared with placebo, mavacamten-treated patients showed greater reductions in post-exercise LVOT gradient and higher peak oxygen consumption after 30 weeks of treatment [40]. Mavacamten has shown short-term palliative improvement of HF symptoms in some patients with obstructive HCM. However, the reduction in outflow gradient was less than with septal myectomy or septal ablation [40]. In addition, the analysis of data from the EXPLORER-HCM trial showed that mavacamten significantly improved overall health and subjective well-being compared with placebo among patients with symptomatic obstructive HCM [41]. Subsequently, the use of mavacamten showed an improvement in a number of cardiopulmonary exercise test parameters beyond pVO2, which indicated a broader benefit in maximal exercise capacity [42]. Importantly, the positive effect of mavacamten in the trial was proven to be independent of beta blocker use in patients [43].

Another study, VALOR-HCM, was designed to determine whether mavacamten allows patients to improve enough that they no longer meet guideline criteria or choose not to undergo ventricular septal reduction therapy (SRT). The first evaluation was conducted after 16 weeks of treatment. It showed that in obstructive HCM patients with untreatable symptoms, mavacamten significantly reduced the fraction of patients meeting guideline criteria for SRT [44]. In addition, it showed a significant reduction in LVOT gradient in patients, as well as an improvement in NYHA functional classification in patients taking mavacamten [44]. Subsequently, the analysis of the study group showed that mavacamten also improved cardiac diastolic function, and this change correlated with improvements in clinical and biomarker endpoints [45]. The second analysis was performed after 32 weeks of randomized clinical trials; unfortunately, the results were not so promising. There was a reduction (13% vs. 10%) in patients requiring or being eligible for STR criteria, but effects were similarly observed in patients who crossed over from placebo after 16 weeks [46].

Summing up, there is some evidence that mavacamten can improve the outcome of patients with HCM. The meta-analysis of randomized clinical trials’ results showed that, in patients with HCM, mavacamten helps improve pVO2, NYHA functional class (improvement by 1), and reduces SRT (with lower rates of SRT or patients who were eligible for SRT). However, the use of mavacamten was associated with higher rates of mild adverse effects, such as dizziness, palpitations, and fatigue. Additionally, there were no significant differences between the groups in the presence of arrhythmia rates [47]. Thus, there are early questions regarding its cost and efficacy. Further research with long-term follow-up is needed to evaluate the efficacy and safety of mavacamten in the treatment of HCM. Nevertheless, according to the 2023 European Society of Cardiology Guidelines, mavacamten should be considered as an adjunct to beta-blockers in adults with resting or provoked LVOTO (Class IIa, Level A) [4].

Recently, another myosin inhibitor, aficamten, has also been studied. Aficamten has a similar mechanism of action to mavacamten, but its half-life and effects on CYP enzymes (which facilitate the oxidation, reduction, and hydrolysis of various molecules such as drugs, toxins, fatty acids, and steroids) are slightly different than those of mavacamten [48]. Aficamten does not induce or inhibit CYP enzymes, unlike mavacamten, which has been shown to induce CYP3A4 and CYP2B6. A phase 2 randomized, placebo-controlled, sequential cohort study, REDWOOD-HCM, showed promising results [49]. These results underscore the potential of sarcomeric targeted therapy in the treatment of HCM. After 10 weeks, the aficamten-treated group had a reduced gradient at rest (mean difference: −40 ± 27 mm Hg and −43 ± 37 mm Hg in cohorts 1 and 2, respectively, *p* = 0.0003 and *p* = 0.0004 vs. placebo) and with Valsalva maneuver (−36 ± 27 mm Hg and −53 ± 44 mm Hg, respectively, *p* = 0.001 and <0.0001 vs. placebo). In addition, aficamten resulted in a reduction in NT-pro BNP levels (by 62% vs. placebo, *p* = 0.0002). The treatment was safe and clinical improvement was observed in those treated with aficamten [49]. Based on these results, a phase 3 trial is currently underway in the multicentered, randomized Sequoia HCM trial, also focusing on symptomatic patients with obstructive HCM (NCT05186818). The trial will complete patient recruitment in September 2023.

### 2.3. Trientine as a New Promising Therapy for HCM

Therapies targeting underlying disease mechanisms in HCM are still limited. As one of the mechanisms in HCM is enhanced oxidative stress, a new randomized clinical trial with the use of trientine (TEMPEST) is now being conducted [50]. Unbound/loosely bound tissue copper II ions are powerful catalysts of oxidative stress and inhibitors of antioxidants. Trientine is a highly selective copper II chelator. In the TEMPEST study, patients with a diagnosis of HCM and in NYHA classes I–III are being randomized to trientine or matching placebo for 52 weeks. The primary outcome will be the change in left ventricular (LV) mass indexed to body surface area, which will be measured using cardiovascular magnetic resonance [50]. The study was planned based on the positive results of a pilot study in which trientine was associated with improvements in cardiac structure and function [51]. In the pilot study, 20 patients with HCM were treated with trientine for 6 months. Additionally, 10 patients with HCM were studied as controls. Trientine treatment was safe and tolerated. Treatment was associated with significant improvements in total atrial strain and global longitudinal LV strain using both echocardiography and cardiovascular magnetic resonance. The left ventricular mass (LVM) decreased significantly in the treatment arm compared with the control group (−4.2 g vs. 1.8 g, *p* = 0.03). A strong trend towards an absolute decrease in LVM was observed in the treatment group (*p* = 0.06) [51]. As a result, Cu^2+^–selective chelation with trientine in a controlled environment was proven to be safe and to be a potential future therapeutic target.

### 2.4. Gene Therapy

Gene therapy in HCM is currently in the research phase, with a focus on genes encoding sarcomeric proteins. Gene editing of the causative gene variants using technologies such as CRISPR/Cas9, gene replacement therapy, and allele-specific silencing are being explored in preclinical studies, but their clinical application is uncertain at this point [52]. Transporting repair material to affected cells is one of the biggest challenges of gene therapy.

Adeno-associated viruses (AAVs) are vectors used in gene therapy for heart disease. AAV serotype 9 (AAV9) has been shown to be a very promising candidate for cardiac gene transfer after systemic administration in mouse and large animal models of HCM [53]. Cardiac gene replacement therapy using AAV has shown promise for HCM-associated myosin-binding protein C3 (MYBPC3) variants in mouse and human-pluripotent-stem-cell-derived cardiomyocytes [54]. In vitro studies have shown that the N-terminal domains (NTDs) of cardiac myosin-binding protein C (cMyBPC) contain regulatory regions essential for sarcomere contractility [55]. Key interactions appear to involve the C0C2 domain of the NTD, which has the greatest effect on Ca^2+^-independent activation of thin filaments [56]. Therefore, recent studies have focused on AAV9 cMyBPC NTD gene transfer containing C0C2 domains (AAV9-C0C2). AAV9-C0C2 vectors were administered to cMyBPC-deficient mice, which significantly improved their cardiac function and reduced histopathological signs of cardiomyopathy and delayed the development of HCM symptoms. Based on these results, human cases with reduced cMyBPC expression may benefit from treatment with C0C2 domains [57]. Moreover, AAV9 was used as a vector in a study on mice with HCM caused by a missense mutation in MYL2 gene encoding for the myosin regulatory light chain (RLC), which is critical for proper cardiac contraction [58,59]. Using AAV9, a phosphomimetic human RLC variant with serine-to-aspartic-acid substitution was delivered into the heart of humanized HCM-D166V mice. Compared with a control group, mice treated with AAV9-S15D-RLC had significantly improved heart performance and increased contractile function [59].

More recently, other techniques using gene-editing technology have also been developed. To date, however, these are isolated studies, albeit ones showing the potential effectiveness of these methods. Further research in this area is needed. The pathogenic MYBPC3 gene was edited using the CRISPR/Cas9 system in a rat model [60]. This was a rat model of HCM (“1098hom”) that carried the Mybpc3 premature termination codon mutation (p.W1098x) discovered in the human HCM pedigree. The 1098hom rats did not express MYBPC3 protein and developed an HCM phenotype (increased ventricular wall thickness and reduced cardiac function). Contrarily, the 1098hom rats injected with a single dose of AAV9 particles, to correct the variant by using a CRISPR genome editing approach, showed after 6 months a restored MYBPC3 protein expression by 2.12%. CRISPR HDR genome editing corrected 3.56% of total mutations and normalized phenotype. The study showed that this is a promising approach for treating HCM associated with MYBPC3 mutation, and CRISPR technology has great potential for treating inherited heart disease [60]. Another study identified an adenine base editor and single-guide RNA system that can effectively correct a pathogenic variant in MYH7 at selected sites in induced pluripotent cardiomyocyte stem cells from HCM patients and in a humanized mouse model of HCM [61]. An overview of gene therapy approaches for HCM is shown in Table 2.

## 3. Fabry Disease (FD)

Fabry disease is a genetic disorder coupled to the X chromosome. Several pathogenic variants in the alpha-galactosidase A (GLA) gene in Xq22 result in insufficient or no lysosomal GLA activity. GLA’s aim is to break down lipids and fats to prevent the accumulation of sphingolipids in blood vessels and tissues. Therefore, a lack of GLA action leads to intracellular accumulation of glycosphingolipids, mainly globotriaosylceramide (Gb3) and its deacylated form, globotriaosylsphingosine (lyso Gb-3). Fabry disease affects many cell types, including endothelial cells, epithelial cells, pericytes, myocardial cells, ganglion cells, and smooth muscle cells [62,63]. Consequently, the main symptoms of FD are cardiac, renal, and nerve damage (peripheral neuropathy) and cardiovascular events. The incidence of classic symptomatic FD is estimated to be between 1 in 40,000 and 117,000 [63]. However, according to the exome sequencing data from 200,643 individuals from the UK Biobank, the overall incidence of variants that cause Fabry disease has been estimated to be 1 in 5573, with the majority associated with a late-onset disease [64].

We can distinguish between two types of FD: classic (early onset) and atypical (late onset). The classic form occurs mainly in men, and the first symptoms appear in childhood, increasing with time. GLA activity <1% indicates classic FD in men [65]. This form is associated with nonsense variants, missense variants, and premature stop variants, resulting in very low or no GLA activity. Atypical FD is more common, and more often undiagnosed, especially in women. In this form, some enzyme activity is usually preserved. Therefore, genetic testing should be performed in suspicion of the atypical form of Fabry disease, rather than just relying on the serum GLA activity assessment [66]. The course of atypical FD is more variable, and symptoms may be limited to a single organ. Patients usually do not have symptoms until age 30 or older. The first sign of FD may be kidney failure or heart disease.

FD is a monogenic disease but shows considerable variability in clinical presentation. This may be due to other genes that can modify the course of FD [67]. A study by Ramirez et al. suggested that there are seven regulatory single-nucleotide polymorphisms in three genes, IL10, TGFB1, and EDN, which can be considered minor modifier genes in FD [68].

In measurements in patients with classic FD, plasma and urine levels of Gb3 and lyso-GB3 are elevated, which could be a potential diagnostic tool for the classic FD [69,70]. Recently, a simple UHPLC-MS/MS method has also been developed to measure Lyso-Gb3 and its analogs from dried blood spots, which may also be able to be used in the diagnosis of FD [71].

### 3.1. Mechanism of Cardiac Involvement in FD

In the course of FD with cardiovascular involvement, cells of the cardiovascular system (i.e., myocytes, endothelial and smooth muscle cells of endocardial vessels, endocardium, valvular fibroblasts, and conducting tissue) accumulate Gb3, which leads to concentric left ventricular hypertrophy with diastolic dysfunction [72,73,74]. Interestingly, the increase in myocardial mass is not solely due to Gb3 accumulation, but rather to the activation of signaling pathways that trigger myocardial hypertrophy and fibrosis [75,76]. The product of Gb3 deacylation, lyso-GB3, also inhibits GLA activity and promotes smooth muscle proliferation, which likely contributes to the thickening of the inner membrane of blood vessels [77,78]. Replacement fibrosis that is observed primarily in posterolateral segments of basal myocardium is a characteristic morphologic feature of Fabry cardiomyopathy. It is also associated with poor outcomes [79,80,81,82]. In addition, about 10% of patients with Fabry disease and LVH develop left ventricular apical aneurysms [83]. Glycolipid accumulation in cardiomyocytes in the conduction structures causes conduction abnormalities from sinus node dysfunction, varying degrees of AV block, and atrial to ventricular arrhythmias [84,85,86].

FD is one of the causes of HCM. Therefore, it is crucial to determine which HCM patients may be affected by FD. Cardiac involvement in FD is characterized by progressive thickening of the heart walls, which can make it difficult to distinguish FD from sarcomeric disease. The incidence of FD in HCM was found to be 0.9%, while the strongest predictors of FD are biventricular block and basolateral late gadolinium enhancement [87]. Moreover, patients with FD present a specific right ventricular deformation pattern in echocardiography, which can also be helpful in the differential diagnosis between the two diseases [88].

### 3.2. Treatment of FD

There are now two approved treatments for FD, enzyme replacement therapy (ERT) and chaperone therapy. However, neither of these methods is a curative treatment for FD. New therapeutic strategies are currently being developed, including second-generation enzyme replacement therapy, substrate reduction therapies, and gene and mRNA therapies.

### 3.3. Enzyme Replacement Therapy (ERT)

ERT is a treatment which replaces an enzyme that is deficient or absent. In FD, ERT is based on an exogenous GLA enzyme supplementation. It is now possible to administer it intravenously every two weeks with agalsidase alfa and agalsidase beta [89]. ERT has been shown to reduce glycolipid substrates, including Gb3, in the urine, plasma, and tissues of FD patients. The therapy improves quality of life, achieving the greatest benefit when started early in the course of the disease [82,90,91,92]. Moreover, long-term observations have shown that ERT is beneficial for LVH including arresting further progression of hypertrophy [93,94,95]. Unfortunately, there are still limited clinical benefits of ERT, due to its short half-life, poor biodistribution, and inability to cross the blood–brain barrier, along with the production of anti-drug antibodies [92,96,97]. It is unclear whether ERT treatment is effective in removing Gb3 from cardiomyocytes. ERT also appears to be ineffective in the later stages of Fabry cardiomyopathy, with fibrosis developed, and it is uncertain whether it slows the progression of fibrosis [82,98,99].

Frustaci et al. showed that ERT had a marked effect on the elimination of Gb3 deposits from enterocytes after two years of treatment with agalsidase alfa. However, cardiomyocytes remained hypertrophied and retained Gb3 deposits. The divergent responses could be explained by different cellular transformations and differences in the expression of mannose-6-phosphate receptor (IGF-II-R), the main carrier of GLA. Enterocytes overexpressed IGF-II-R, while cardiomyocytes reduced it 7-fold, compared to healthy controls [100]. Additionally, the authors found that cardiomyocytes, compared to enterocytes, had a decreased amount of IGF-II-R [100]. Additionally, another study was conducted with frozen endomyocardial biopsy samples from seventeen FD patients who underwent at least three years of ERT therapy. The results showed a 5.4-fold decrease in IGF-II-R in comparison to the normal hearts. [101].

### 3.4. Second Generation ERT

The second generation of ERT includes enzyme therapies that have been biotechnologically enhanced to improve their uptake by cells and increase their efficacy. Pegunigalsidase alfa is a newly developed pegylated form of α-GAL. Due to the pegylation of the enzyme and epitope masking, it is less immunogenic. In GLA-deprived mice, the compound had better stability and longer half-life in the circulation than conventional ERT, with a significant reduction in Gb3 content in the kidney and heart [102]. In three studies (BRIDGE, BRIGHT, and BALANCE), an intravenous dose of 2 mg/kg administered every four weeks was tested for safety, efficacy, and pharmacokinetics. The effect was compared with the use of algasidase alfa and beta [103]. The results of these studies allowed for the approval of pegunigalsidase alfa (PRX-102, pegunigalsidase alfa-iwxj) as a second-generation ERT for the treatment of Fabry disease by the European Commission (5 May 2023).

### 3.5. Chaperone Therapy

Chaperone therapy (migalastat), which involves oral administration of iminosugar, is another form of FD treatment. It has been approved for patients with FD aged ≥ 12 years and with amenable GLA variants [104]. In these patients, migalastat restores normal folding and stability of the mutant enzyme, resulting in a reduction in Gb3 and its substrates [105]. Migalastat treatment has several advantages over ERT: oral administration, non-immunogenic nature, better sustained enzyme activity, ability to cross the blood–brain barrier, and potentially better biodistribution. In addition, it has been shown to be safe, has high patient compliance, and the incidence of severe clinical complications of Fabry disease in patients during long-term treatment with migalastat was low [106,107,108]. Long-term follow-up and randomized trials have shown that chaperone therapy reduces LV hypertrophy, and left ventricular mass index decreased significantly from baseline (−7.7 g per square meter; 95% CI, −15.4 to −0.01) [106,109,110]. In addition, 18-month treatment stabilized LV mass and improved exercise tolerance in untreated FD patients with cardiac involvement [111]. One of the biggest drawbacks of oral chaperone therapy is that it is only effective in patients with amendable GLA variants, which accounts for 35–50% of patients [109]. However, the response to migalastat varies from patient to patient, due in part to the large range of increases in GLA activity after treatment. So far, the efficacy of migalastat has been comparable to ERT in reducing the overall progression of the disease.

### 3.6. Substrate Reduction Therapy (SRT)—Lucerastat, Venglustat

Substrate-reducing therapy is designed to inhibit Gb3 accumulation in cells, and its use in combination with ERT may have potential as an adjunctive treatment [112,113]. SRTs are orally administered iminosugars that directly inhibit glycosphingolipid synthesis, thereby lowering the cellular burden of Gb3. Studies using fibroblasts from FD patients have shown a reduction in Gb3 after SRT use, as well as improvements in abnormal cell membranes [113,114,115]. Lucerastat, by reducing the net load of Gb3 in tissues, has a disease-modifying potential to improve symptoms and delay disease progression [116]. The phase 3 of a clinical trial in which lucerastat will be administered to patients with FD (MODIFY, NCT03425539) is ongoing, with the enrollment of 118 participants. However, its results have not yet been published. Another SRT is venglustat, which inhibits the enzymatic conversion of ceramide to glucosylceramide, thereby reducing the amount of substrate available for the synthesis of more complex glycosphingolipids. Venglustat has a proven favorable safety and tolerability profile [117]. Subsequently, an open-label, single-arm, uncontrolled 26-week phase 2a clinical trial (NCT02228460) and a 130-week extension study (NCT02489344) were conducted to evaluate the safety, pharmacodynamics, pharmacokinetics, and exploratory efficacy of oral venglustat 15 mg once daily in untreated adult men with classic Fabry disease. As an effect, with long-term treatment with venglustat, regression of Gb3 accumulation in superficial skin capillaries was observed in adult men untreated with ERT with classic Fabry disease [118].

### 3.7. Gene Therapy

Gene therapy is considered potentially more effective than ERT due to its ability to induce endogenous production of GLA. The enzyme undergoes patient-specific post-translational modifications resulting in a greater uptake of the enzyme in heart and kidney tissues in patients with Fabry disease. Therefore, a study was initiated on a mouse model of FD using plasmid DNA to deliver the GLA sequence in a vector of solid lipid nanoparticles (SNLs) functionalized with galactomannan [119]. The study achieved clinically relevant levels of α-Gal A in plasma, liver, and, most importantly, heart and kidney. In addition, the vector showed no significant erythrocyte agglutination or hemolytic activity, which suggested its safety. Subsequent studies have used adenovirus as a vector. Preclinical in vivo studies using liver-targeted adenovirus-mediated gene transfer in a mouse model of FD showed a dramatic increase in α-Gal A activity and a significant reduction in lyso-Gb3 [120]. It is unclear whether the released enzyme will result in adequate uptake by affected tissues.

GLA genes can also be transferred into liver cells via adenoviral vector (AAV). The FLT190 is a novel, potent, engineered capsid (AAVS3) that carries a codon-optimized human GLA cDNA under the control of the liver-specific promoter FRE1 [121]. In primary human hepatocyte cell cultures, AAVS3 showed superior transduction efficiency compared with AAV5 and AAV8 variants currently used in the clinic [122]. However, FLT190 genome was pseudo-typed with AAV8 for efficient transduction in mice models. Fabry mice treated with AAV8-FLT190 showed an increased level of GLA in plasma and affected tissues as well as decreased storage of Gb3 and lyso-Gb3 in the plasma, urine, kidney, and heart [121]. In non-human primates (juvenile rhesus), GLA was also increased in plasma after administration of FLT190, and no FLT190-related toxicities or adverse events were observed. Promising results led to further gene therapy research in a phase 1/2 clinical trial in patients with Fabry disease (NCT04040049), evaluating further potential benefits of FLT190. However, no results are posted yet, and till the middle of 2023, three patients have been enrolled.

Another approach uses cardiotropic vectors that can specifically target myocardial tissue with increased delivery and reduced immunogenicity, and which are currently being tested in primates. The STAAR clinical trial (NCT04046224) is currently underway, which is the first human study using ST-920, a recombinant AAV2/6 vector encoding the cDNA for human GLA. This study is evaluating the safety and tolerability of escalating doses of ST-920 to ensure stable and effective long-term production of α-Gal A that reaches therapeutic levels in patients with FD.

Finally, initial experiments with gene delivery systems are still being developed. Encapsulation of human GLA mRNA in lipid nanoparticles increased GLA levels in liver, heart, and kidney in mice and primates [123]. This provides a basis for developing further therapies in FD using this technology. The summary of the gene therapy approaches for FD has been presented in the Table 3.

## 4. Pompe Disease (PD)

Pompe disease is a severe metabolic myopathy whose symptoms are caused by pathogenic variants in the gene encoding acid alfa-glucosidase (GAA). The function of this enzyme is to break down glycogen in lysosomes. Lack of the enzyme leads to lysosomal glycogen accumulation in many tissues. However, in PD, symptoms and dysfunction mainly affect the heart muscle and skeletal muscle.

There are two types of the disease, depending on the time of onset of symptoms and the presence or absence of cardiomyopathy. Classical Pompe disease of infantile onset (IOPD) is characterized by a more severe clinical course, age of onset of symptoms ≤ 12 months, rapidly progressive hypertrophic cardiomyopathy, left ventricular outflow obstruction, and respiratory muscle weakness leading to respiratory failure. Untreated patients usually die within the first year of life [124,125]. Patients with a similar clinical picture in the first year of life, but with cardiomyopathy without left ventricular outflow obstruction, are referred to as having non-classical IOPD [126]. When the disease manifests after the age of 12 months, it is classified as late-onset Pompe disease (LOPD), which has a much slower and milder course. To make the diagnosis, it is necessary to demonstrate a deficiency in GAA enzymatic activity. Currently, this test is often included in newborn screening. GAA mutation analysis is also a standard. In doubtful cases (especially in LOPD), a muscle biopsy can be performed.

PD has a very broad clinical spectrum, with patients varying widely in terms of symptoms, age of onset, and severity of organ involvement. The clinical course depends on the type of pathogenic variants and the resulting level of residual GAA activity. Some studies have suggested a potential role for angiotensin-converting enzyme (ACE) polymorphism in modulating clinical outcome in patients with LOPD [127]. However, the insertion/deletion (I/D) polymorphism of the ACE was later found not to explain the large differences in disease severity and response to ERT observed among Pompe disease patients with the same c.-32-13T>G GAA variant [128]. Mutations in the GAA gene, located on chromosome 17 q25 [129], affect various stages of the enzyme (such as protein synthesis, post-translational modifications, lysosomal transport, and maturation). Some mutations can be population specific; e.g., c.-32-13T>G (IVS1) is the most common defect in Caucasians. This mutation preserves low levels of the normal enzyme [130,131,132]. In addition, two sequence variants, c.1726G>A and c.2065G>A, are known to cause so-called pseudo-deficiency—despite low levels of GAA activity, patients do not show clinical symptoms, and it leads to false positives in newborn screening tests [133,134].

### 4.1. Enzyme Replacement Therapy with Alglucosidase Alfa

ERT is now the standard therapy in PD. Alglucosidase alfa was first approved in 2006, and its efficacy has been demonstrated in several clinical trials. In the first clinical trial in infants, ERT significantly improved survival, reduced the risk of death (by 99%), and the risk of invasive ventilation (by 92%) compared to the results of untreated subjects [135]. However, a longer follow-up showed that, although ERT prolonged survival, its rate decreased significantly (up to 65%) in the first years of life of children with PD, and the number of patients who became ventilator-dependent increased up to 30–58% [136,137,138,139]. The therapy also improved cardiovascular parameters with a significant reduction in left ventricular mass index and left ventricular wall thickness, correction of abnormal ECG parameters, and improvement in cardiac function. Likewise, many patients reached major milestones in motor development. A study published in 2023, involving 27 patients treated with ERT, showed that cardiac function (measured by echocardiography) normalizes after starting ERT and remains stable for a median follow-up period of 9.9 years [140]. Recently, some studies have suggested that the currently registered dosage should be reconsidered. Improved outcomes and better survival were demonstrated in 4 patients receiving higher and more frequent doses of the drug (40 mg/kg/week) instead of the currently recommended 20 mg/kg every other week [141,142]. Moreover, patients with classic pediatric PD treated with the high ERT dose of 40 mg/kg per week had significantly better survival compared to patients treated with the standard recommended ERT dose of 20 mg/kg every other week [142].

To date, there has been only one randomized, double-blind, placebo-controlled phase 3 clinical trial of alglucosidase alfa for the treatment of LOPD in children and adults, and several observational studies [143,144,145,146,147,148,149]. These showed improvements in patients treated with alglucosidase alfa in terms of improved gait distance and stabilization of respiratory function. However, the variability in response to treatment was very high. The most reliable effect of ERT was improvement in cardiac function, regardless of the severity of the disease. A later analysis in the ADVANCE trial evaluated the effect of 52-week alglucosidase alfa treatment on cardiac function. It showed that 4000 L alpha-glucosidase therapy maintained fractional shortening, left ventricular end-diastolic and septal thickness, and improved left ventricular mass score [150]. In contrast to cardiac function, skeletal muscle response is more variable. A recent metanalysis included 16 studies with data from 589 patients with LOPD and analyzed 419 patients. The available data showed that ERT had a significant beneficial effect on improving walking distance (32.2 m more during the follow-up period; *p* = 0.0003) in patients with LOPD and a non-significant improvement in muscle strength. There was also no improvement in volumetric functional capacity [151].

The therapy is now more available for patients with PD. Interestingly, Dutch analyses suggest that home infusions of alglucosidase alfa ERT can be used safely. In the study, 116 of 120 eligible patients (17 with classic childhood onset, 2 with atypical childhood onset, 15 with childhood onset, and 82 adults) completed 423 safety questionnaires (response rate: 88.1%). Symptoms during or after infusion were reported 27 times in 17 patients with the most common being fatigue (in 9.5% of patients) [152]. No event requiring hospitalization was reported [152]. Moreover, the first reported use of ERT in a fetus with IOPD was proven safe and effective. The family history was positive for IOPD with cardiomyopathy—the disease had been diagnosed in two previously deceased siblings. After receiving ERT in utero and standard postnatal therapy, the patient had normal heart function and age-appropriate motor function at birth. The patient also reached developmental milestones, had normal biomarker levels, was eating well, and was growing at 13 months [153].

### 4.2. Therapies Designed to Enhance the Effect of ERT

Although ERT has proven effective, it has some drawbacks and limitations. Brain magnetic resonance imaging revealed abnormalities in the brain’s white matter in long-term ERT-treated patients with classic IOPD [154]. These data indicate a limitation of ERT—the inability of the recombinant enzyme to cross the blood–brain barrier. Another drawback of ERT is the development of antibodies to the exogenous protein in almost all patients with Pompe disease [155,156]. Moreover, the effect of alglucoside-alfa is variable, and some patients do not respond as well and do not show a sustained benefit of therapy. Thus, there is still a need for improved ERT or new treatment strategies for PD. One of these is avalglucosidase alfa, a recombinant human GAA enzyme replacement therapy. It is designed to enhance enzyme delivery by increasing mannose-6-phosphate receptor targeting and enzyme uptake to increase glycogen clearance. Avalglucoside alfa (Neo-GAA) is a second-generation alglucosidase. Its efficacy was first demonstrated in preclinical studies (reducing glycogen to similar levels as alglucoside alfa at a much lower dose), and its safety was proven in a phase 1 clinical trial [157,158]. Subsequently, the phase 2 Mini-COMET study showed that avalglucosidase alfa in patients with IOPD who had previously failed to show a positive effect of alglucosidase alfa responded after 25 weeks of avalglucosidase alfa [159]. The recently published COMET trial (a phase 3 study) included participants who had not been previously treated with ERT [160]. Patients treated with avalglucosidase alfa and alglucosidase alfa were observed for 49 weeks. It provided evidence of clinically significant improvements in respiratory function, mobility, and functional endurance with avalglucosidase therapy compared to alglucosidase alfa therapy. No serious adverse events were reported [160]. Treatment with avalglucosidase alfa resulted in a least-squares mean improvement in upright forced vital capacity (FVC%) predicted of 2.89% (SE 0.88) compared with 0.46% (0.93) with alglucosidase alfa at week 49 (difference 2.43% [95% CI –0.13 to 4.99]). Improvements were also seen in the six-minute walk test (6MWT) with avalglucosidase alfa compared with alglucosidase alfa, with greater increases in distance covered (difference 30.01 m [95% CI 1.33 to 58.69]) and percent predicted (4.71% [0.25 to 9.17]) [160]. Further reports on the results of avalglucosidase alfa treatment are presented in the study extension. After 49 weeks of the COMET study, all patients received 20 mg/kg of avalglucosidase alfa every other week. Efficacy was assessed after 97 weeks, and safety was assessed until the last follow-up. Of the 100 participants from the double-blind treatment period, 95 went on to the extension period. This randomized extension of the clinical trial showed maintenance of positive clinical outcomes in patients continuing avalglucosidase alfa treatment and, to a lesser extent, in patients who switched from primary alglucosidase treatment. No new safety risks were observed [161].

Another approach is to increase efficiency in a mouse model by up-regulating action-independent mannose 6-phosphate receptor (M6PR) Cl-MPR [162]. An attractive experimental approach involves grafting a synthetic analogue of M6P onto recombinant human acid alfa-glucosidase (rhGAA), leading to a significant increase in the affinity of the recombinant enzyme for M6PR without changes in catalytic activity. This modified enzyme significantly improved muscle function even in difficult-to-treat old knock-out (KO) mice, while rhGAA was inactive [163]. Regulation of the CI-MPR receptor by the β2-agonist clenbuterol or albuterol has been shown to enhance ERT efficacy in a mouse model [164,165].

A pilot open-label study of adjunctive albuterol therapy in patients with LOPD who did not improve after ERT demonstrated the safety and potential efficacy of this therapy [166]. The potential benefits of albuterol have also been demonstrated in carefully selected patients with late-onset Pompe disease in a 24-week, double-blind, randomized, placebo-controlled phase 1/2 study [167]. In the albuterol group, forced vital capacity in the supine position increased by 10% (*p* < 0.005), and forced expiratory volume in one second increased by 8% (*p* < 0.05); the six-minute walk test increased by 25 m (*p* < 0.05; excluding one participant who was unable to complete the muscle function test) [167]. Also, there is now the first evidence supporting the efficacy of clebuterol from a 52-week, randomized, double-blind, placebo-controlled phase 1/2 study [168]. Improvements in 6MWT and rapid motor function test scores were observed in subjects treated with supportive clebuterol [168]. In addition, clenbuterol reduced the glycogen content of the vastus lateralis muscle by 50% at week 52.

Another promising treatment method is chaperone therapy. Chaperones promote the folding, stability, and lysosomal transport of administered enzymes in therapy. This improvement in the stability of alglucosidase alfa in the blood has been observed in patients receiving ERT in combination with the iminosugar N-butyldeoxynojirimycin [169]. A similar approach was used combining the chaperone iminosugar miglustat (also known as AT2221, enzyme stabilizer) with ATB200 (cipaglucosidase alfa, a novel recombinant human acid α-glucosidase with high levels of mannose-6-phosphate). Initially, this effect was shown in a mouse model of PD when ATB200 was administered in combination with the pharmacological chaperone AT2221 (miglustat), which stabilizes the enzyme and improves its pharmacokinetic properties [170]. ATB200/AT2221 was significantly more potent than alglucosidase alfa. The drug dramatically reduced the accumulation of autophagy, a major secondary defect in an affected muscle. Reversal of lysosomal and autophagic pathologies led to improved muscle function. These data demonstrated the superiority of ATB200/AT2221 over currently approved ERT in a mouse model. However, the data derived from the PROPOL study revealed that the treatment with cipaglucosidase alfa plus miglustat did not achieve statistical superiority over alglucosidase alfa plus placebo in improving the 6 min walking distance in a general population of patients with the late-onset Pompe disease [171]. Now, there is a clinical trial which aims to find out if the co-administration of investigational ATB200 and AT2221 is safe in adults with Pompe disease (NCT02675465). No results have been published yet as the trial is expected to be completed in December 2023.

### 4.3. Gene Therapy Strategies

A potential alternative to ERT is gene therapy. Compared to ERT, gene therapy may offer several advantages to PD patients. Importantly, gene therapy could potentially be more effective than ERT, especially if administered early in the course of disease progression. This increased efficacy could be due to its potential ability to cross the blood–brain barrier as demonstrated in the correction of neurological symptoms associated with mucopolysaccharidosis IIIB in mice following the use of certain AAV serotypes [172].

Initial studies using adenoviruses (Ad), adeno-associated viruses (AAV), and retroviruses have demonstrated the feasibility of gene therapy in PD [173,174,175]. Several studies have been conducted in mouse models, ranging from direct injections of recombinant adeno-associated virus (rAAV) expressing the GAA protein [175,176], to intralingual administration of AAV that resulted in temporary correction of motoneuron pathology in KO mice [177], to systemic and intrathecal administration of rAAV1-hGAA that showed improved respiratory function in mice [178,179]. Based on these preclinical studies, the first human trial of diaphragmatic gene therapy (AAV1-CMV-GAA) was conducted in five children with IOPD who required assisted ventilation prior to the study [180,181,182,183]. The study demonstrated the safety of the AAV treatment but no clinical outcome. An additional clinical trial has been conducted; unfortunately, the results have not yet been published (NCT02240407). The study will use a recombinant AAV vector carrying a codon-optimized acid alpha-glucosidase gene under the control of the human desmin promoter (rAAV2/9-DES-hGAA) [184]. Since desmin is highly expressed in a muscle, this improves the expression level of the GAA transgene. In studies with a mouse model of PD, neuronal and cardiorespiratory functions improved after systemic or intrapleural administration of this vector [185,186].

Another approach is gene therapy using the liver’s high metabolic capacity to produce and secrete GAA protein. This strategy is based on the use of liver-tropic AAV8. Systemic injection of a modified AAV8 vector containing a liver-specific promoter (AAV2/8-LSPhGAA) induced immune tolerance to rhGAA and improved ERT efficacy in a mouse model of PD [187]. Preclinical studies have shown that secreted GAA is taken up by cardiac and skeletal muscles, leading to glycogen reduction and improved muscle function [188]. In a more recent study, the authors demonstrated with in vivo experiments that transgenes, delivered by the hepatic tropic rAAV, resulted in high levels of secreted GAA, low immunogenicity, and metabolic changes in muscle, central nervous system, and spinal cord. So, it has been shown that liver-secreted GAA is efficiently utilized by peripheral tissues [189].

Recently, the first results of a phase 1 study of gene therapy using a vector (AAV8-LSPhGAA) in humans have been published [190]. Following the first dose in three patients with LOPD, the initial safety and bioactivity of the first dose was noted at a 52-week follow-up (NCT03533673). After week 26, based on the detection of elevated serum GAA activity and no clinically significant decreases according to the study protocol, the patients discontinued their ERT. All patients showed sustained serum GAA activity between 101% and 235% of the baseline minimum activity two weeks after the previous ERT dose. No treatment-related serious adverse events occurred. At week 52, GAA activity in the cohort’s muscles was significantly increased (*p* < 0.05). These data confirm the safety and bioactivity of AAV8-LSPhGAA, the safety of ERT withdrawal, and the effective immunoprophylaxis, and they justify further clinical development of AAV8-LSPhGAA therapy in Pompe disease [190]. The summary of gene therapy strategies for PD has been presented in the Table 4.

## 5. Danon Disease (DD)

Danon disease is a rare X-chromosome-associated disorder that is caused by mutations in the gene encoding lysosomal membrane protein 2 (LAMP2). LAMP-s deficiency results in a lack of lysosome binding to autophagosomes, leading to abnormalities in autophagy, and thus to glycogen accumulation. Typically, men are more affected than women, who appear to have a broader and more variable phenotype with predominant myocardial involvement [191,192]. The onset of the disease always manifests on the cardiac side, earlier in men (in childhood, with a median age of 12 years) than in women (in adolescence or adulthood with slower disease progression than in men). 

Men with DD usually have a triad of symptoms: severe cardiomyopathy, myopathy (symptoms of muscle weakness in 80–90%), and mild intellectual disability (cognitive impairment in about 70%) [193]. HCM occurs in 70–88% of men with DD [192,194]. The disease has a rapidly progressive, massive biventricular hypertrophy that leads to end-stage heart failure; therefore, patients often undergo heart transplantation (HTx). On the other hand, only about 18% of female patients require HTx, compared to almost all male patients [193]. Rarely, HCM can evolve into dilated cardiomyopathy (DCM), which is seen more often in women [193]. The cardiomyopathy affects more than 70% of women with DD, with dilated cardiomyopathy occurring in 30–50% of them and hypertrophic cardiomyopathy in the rest [192,194]. Patients with DD have a high incidence of life-threatening arrhythmias. Signs of preexcitation, including Wolff–Parkinson–White (WPW) syndrome on the ECG, occur in more than 80% of affected men [193]. The source of preexcitation (and thus accessory pathways) may be the disruption of the fibrous ring by glycogen-filled myocytes [195,196].

The prevalence of DD is very low, and the exact number has not been determined. However, genetic studies of patients with HCM have shown that the prevalence of individuals with a pathogenic LAMP2 variant in this group is 1–4% [197,198]. Next-generation sequencing screening of LAMP2 gene mutations identified an increasing number of patients with DD; their prevalence is currently estimated at 4–6% among children with HCM [198,199,200] and about 0.7–4% among adults with HCM [197,201,202].

The diagnosis of DD is established by identifying a pathogenic variant in LAMP2 by molecular genetic testing (hemizygous in males and heterozygous in females). Muscle biopsy shows vacuolar myopathy and significant fibrosis. The pathological hallmark of the disease is the presence of cytoplasmic vacuoles and/or granules showing increased activity of lysosomal enzymes, including acid phosphatase and nonspecific esterases [203]. Tissue staining showing the absence of LAMP-2 protein can confirm the diagnosis, but it is not required in all of the Danon disease cases.

### 5.1. Genotype–Phenotype Correlations

To date, specific genotype–phenotype correlations for pathogenic loss-of-function variants have not been confirmed. A small number of pathogenic missense variants restricted to exon 9B of the LAMP-2B protein isoform have been reported in the literature and have been associated with a milder or atypical DD phenotype (e.g., mild cardiomyopathy and cognitive impairment with severe myopathy, muscle weakness and mild left ventricular dysfunction, and a patient with myopathy and hepatopathy) [193,204,205,206]. Null mutations were associated with the earliest age of onset, splicing mutations showed an intermediate age of onset, while missense mutations showed the latest age of onset in male patients [193]. Recently, a new pathogenic splicing altering mutation in the LAMP2 gene (c.741+2T>C), with only cardiac involvement, was described. [207].

LAMP-2 neither has enzymatic activity, nor does it directly affect glycogen storage or degradation. Its role is to mediate macromolecule transport and fusion of lysosomes with endosomes, phagosomes, and the plasma membrane. Its three isoforms, LAMP-2A, LAMP-2B, and LAMP-2C, are formed by alternative splicing of the terminal exon 9. Most pathogenic variants result in the absence of all three isoforms of LAMP-2 protein. So far, isoform-specific mutations have been found only in LAMP-2b, suggesting that LAMP-2b deficiency is sufficient to cause disease [208]. Studies in an animal model of DD and pluripotent stem cells suggest that the absence of the LAMP-2B isoform causes impaired macrophagy and subsequent mitochondrial dysfunction [209,210,211].

With a targeted NGS, it is possible to detect pathogenic variants of the LAMP2 gene in any patient with high sensitivity. At least 110 different variants in the LAMP2 gene are associated with DD. The most common of these is the c.926G>A mutation (causing exon 7 to be skipped) [212]. Most were null mutations (nonsense mutations, deletions/insertions with frameshift) or splicing mutations. A synonymous substitution (c.864G>A) resulted in exon 6 skipping, indicating that seemingly silent changes can be deleterious when located near splicing sites [213]. The deleterious effect of null mutations occurs because premature termination codons lead to the synthesis of truncated, non-functional proteins that can be dangerous, and their synthesis is inhibited by the mRNA degradation mechanism in the nucleus [214]. Recently, a novel splice site variant (NM_013995.2:c.864+5G>A) located in intron 6 of the LAMP2 gene was identified [215]. The pattern of X chromosome inactivation (XCI) found in female cells is associated with clinical heterogeneity in women with DD. When random XCI occurs, LAMP-2 expression in skeletal muscle fibers may be preserved, but not in non-regenerating cardiomyocytes. This explains why most patients developed cardiomyopathy but not skeletal myopathy [212].

### 5.2. Prospects for Therapies

Treatment of patients with DD is currently based on treating heart failure and arrhythmias and preventing SCD. Ultimately, the disease in its final stage leads to HTx. One active observational clinical trial is currently underway to characterize the natural history of DD and the results of HTx in this group. 

The first clinical trial of gene therapy in DD began in April 2019. It is a non-randomized, open-label, phase 1 study to evaluate the safety and toxicity of gene therapy using a recombinant adenoviral virus containing the human LAMP2b trans isoform gene (RP-A501). The study is planned for 12- to 24-year-old male DD patients divided into four cohorts (8–14 years or >15 years) who will receive two different doses of RP-A501. The aim of the study is to evaluate the safety and whether RP-A501 induces cardiomyocyte and skeletal muscle transduction along with gene expression. In addition, a preliminary assessment of the degree of histological correction of cardiomyocytes by endomyocardial biopsy will be performed. Likewise, a preliminary assessment of clinical stability will be conducted. The preliminary results have thus far shown that the treatment has been generally well tolerated and has led to cardiac LAMP2B gene expression associated with a provisional evidence of cardiac and extra-cardiac benefits [216]. The full results of the study are still awaited. The therapy is based on the results of studies in mouse models, which showed that mice receiving AAV9. LAMP2B showed dose-dependent restoration of human LAMP2B protein in heart, liver, and skeletal muscle tissue [217]. Moreover, cardiac function was also improved and transaminases were reduced in KO mice treated with AAV9.LAMP2B, indicating a beneficial effect on the heart and liver [216]. Survival of aged mice treated with gene therapy was also clearly improved [216].

## 6. Conclusions

In conclusion, this article provides insights into the genetics, pathogenesis, treatment options, and potential therapies for hypertrophic cardiomyopathy and related cardiomyopathies. The differentiation between HCM, Pompe disease, Fabry disease, and Danon disease is crucial, since the therapy should treat the cause, not just the symptoms, of heart failure. The substitutional therapy, however, is insufficient in diseases caused by enzyme deficiencies. This underscores the need for continued research and therapeutic development to improve outcomes for patients with these conditions. Some first results of gene therapy seem promising.

## Figures and Tables

**Table 1 ijms-24-13239-t001:** Comparison of Genetic Disorders Affecting the Heart: Hypertrophic Cardiomyopathy, Pompe Disease, Danon Disease, and Fabry Disease.

	Hypertrophic Cardiomyopathy	Pompe Disease	Danon Disease	Fabry Disease
**Genetic Basis**	Pathogenic variants in genes encoding sarcomere proteins.	Pathogenic variants in gene encoding acid alfa-glucosidase (GAA).	Pathogenic variants in gene encoding lysosomal membrane protein 2 (LAMP2).	Pathogenic variants in gene encoding alpha-galactosidase A (GLA).
**Clinical Manifestations**	Heart failure symptoms and arrhythmias.	Cardiac and skeletal muscle involvement.	Cardiomyopathy and myopathy, intellectual disability in men.	Multi-systemic involvement, cardiovascular, renal, neurological symptoms.
**Diagnostic Methods**	Echocardiography, cardiac magnetic resonance, genetic testing.	Deficiency in GAA enzymatic activity, mutation analysis, muscle biopsy.	Genetic testing for LAMP2 variants, muscle biopsy.	Blood test for alpha-galactosidase A (GLA) activity, GLA pathogenic variants analysis.

**Table 2 ijms-24-13239-t002:** Gene Therapy Approaches for Hypertrophic Cardiomyopathy (HCM).

Approach	Target	Vector	Model	Findings/Outcomes
**AAV9 cMyBPC NTD Gene Transfer** **[57]**	Cardiac Myosin-Binding Protein C (cMyBPC)	AAV9	cMyBPC-deficient mice	Improved cardiac function, reduced histopathological signs of cardiomyopathy, delayed HCM development.
**AAV9-S15D-RLC Gene Transfer** **[59]**	Myosin Regulatory Light Chain (RLC) D166V mutation	AAV9	Humanized HCM-D166V mice	Improved heart performance, increased contractile function.
**CRISPR/Cas9 Editing of MYBPC3** **[60]**	Myosin-Binding Protein C3 (MYBPC3)	AAV9	Rat model (1098hom)	Restored MYBPC3 expression by 2.12%, CRISPR HDR genome editing corrected 3.56% of total mutations and normalized phenotype.
**Adenine Base Editor and sgRNA System** **[61]**	Myosin Heavy Chain 7 (MYH7)	AAV9 + single plasmid + sgRNA	HCM patient-derived cells, humanized mouse model of HCM	Normalization of contractile force and rescued cellular energetics.

**Table 3 ijms-24-13239-t003:** Gene Therapy Approaches for Fabry Disease: Description and Clinical Trials.

Gene Therapy Approach	Description	Ongoing Clinical Trials
**Plasmid DNA via Solid Lipid Nanoparticles**	Plasmid DNA delivering GLA via lipid nanoparticles in mice. Achieved clinically relevant α-Gal A levels in plasma, liver, heart, and kidney.	-
**Adenovirus Vector**	Adenovirus-based gene transfer in mice. Dramatic increase in α-Gal A activity, significant reduction in lyso-Gb3.	-
**AAV Vector (FLT190)**	AAV carrying GLA cDNA in mice and primates. Increased plasma GLA, reduced Gb3 and lyso-Gb3. Positive results in animals.	Phase 1/2 trial ongoing (NCT04040049).
**Cardiotropic AAV Vectors (e.g., ST-920)**	A recombinant AAV2/6 vector encoding the cDNA for human a-Gal A.	First in human trial—ongoing trial (NCT04046224).
**Encapsulation of Human GLA mRNA**	Lipid nanoparticles in mice and primates. Increased GLA in liver, heart, and kidney. Promising increase in enzyme levels.	-

**Table 4 ijms-24-13239-t004:** Gene Therapy Strategies for Pompe Disease.

Gene Therapy Strategies	Method	Studies	Human Trials
**AAV-based Gene Therapy**	Uses AAV vectors	Mouse models: Improved respiratory function	Phase 1 trials ongoing (NCT02240407)
**Liver-targeted Gene Therapy**	Uses liver-tropic AAV8 vectors	Mouse models: Improved ERT efficacy	Phase 1 trial results published [190]

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
