# Peer review of "Hypertrophic Cardiomyopathy versus Storage Diseases with Myocardial Involvement"

_ijms, 2023, doi:10.3390/ijms241713239_

Round 1
Reviewer 1 Report
Burban et al. Hypertrophic cardiomyopathy versus storage diseases with myocardial involvement
Overall: this paper is well-written and quite comprehensive. The paper is organized around HCM due to sarcomeric variants and 3 other genocopies (Fabry, Pompe, Danon). Other causes of HCM are not addressed, and the reviewer notes that to be complete, perhaps they should be (Freidreich ataxia, Noonan/RaASopathies, PRKAG2, TTR Amyloidosis). The authors say they want to focus on storage disorders but a clear explanation for why some HCM conditions are not covered is hard to find.
There are many HCM genetics reviews published and so this does not necessarily add to the overall literature. However, the article is a nice balance of genetics, mechanisms, phenotypes, and current and emerging therapies.
Minor:
Table 1: indicates that HCM patients due to sarcomeric variants have a “Wide spectrum of cardiac and skeletal muscle symptoms” which is perhaps not accurate as skeletal muscle symptoms are infrequent I this group (and the authors do not describe any such skeletal features in detail in the text).
Line 322: the enzyme assay for Fabry disease is unreliable in females and this should be mentioned. To be more technically correct this is a serum enzyme activity assay (not just a ‘blood test’)
Line 390: recently pegunigalsidase alfa-iwxj has been approved as a 2nd generation ERT for Fabry disease
Article is written (introduction) in first person (‘we’); which is perhaps a bit less typical for a review
Line 80: LVOTO is not defined (LVOT is defined above)
Line 95-96: “Those “sarcomere positive” have an identifiable pathogenic or likely pathogenic genetic variant and mutations in more” [meaning of ‘more’ and logic of this sentence not clear]
The term ‘mutation(s)’ and ‘pathogenic variants’ are both used; current preference in many publications is to use pathogenic/likely pathogenic and not the ‘mutation’ terminology. [authors may consider adjusting paper to focus on ‘variants’ rather than ‘mutations’]
Reviewer 2 Report
The Review by Burban et al. describes the genetics and pathogenesis of HCM, Fabry disease, Pompe disease and Danon disease. Current and experimental treatment options, such as pharmacological intervention and gene therapies were also reported.
Major comments:
1) Are the Clinical Manifestations and Diagnostic Methods reported in Table 1 the more appropriate for HCM? I would have some doubts on the skeletal muscle symptoms related to HCM and the presence of muscle biopsy as a diagnostic method instead of echocardiography.
2) The reference 15 is: “COMMENT ESC working group.” I do not know if this can be accepted as a reference. Please cite the most recent Review on the HCM genetics and if additional genes were identified after the publication of the cited Review, add the new publications.
3) Line 99: “Clinical heterogeneity and different phenotypic expression…” are basically the same concept. Perhaps two important aspects of HCM should be pointed out here: clinical heterogeneity and incomplete/reduced penetrance.
4) The phrase at lines 136-137 “However, it was recently shown that only 38.5% of patients diagnosed with HCM carrying an HCM-associated variant in MYBPC3 or MYH7 had a clinical genetic test result” needs to be better explain. The authors have to explain the “Genome-first” approach used in the paper cited [ref. 31], otherwise the readers cannot understand the meaning of the phrase, that seems out of the context.
5) The sentence at lines 145_146 “Recently, rare minor protein-modifying variants (termed pathogenic in HCM) manifesting some unique phenotypic features have been identified” is not entirely clear. Is it related to the next phrase? What proteins is it referring to? Please explain in more detail the specific proteins involved and their unique phenotypic features. Please add references as well.
6) At line 211 it is reported “However, the use of mavacamten may be associated with more adverse effects.” The authors should explain the adverse effects in more details citing the corresponding references.
7) In the paragraph “a. Gene therapy” the authors report a paper on a rat model of HCM ("1098hom") carrying the Mybpc3 premature termination codon mutation. In this paper a CRISPR/Cas9 approach was used to rescue the phenotype of the rat. In fact, the 1098hom rats didn't express MYBPC3 protein and developed an HCM phenotype with increased ventricular wall thickness and diminished cardiac function. On the contrary the 1098hom rats injected with a single dose of AAV9 particles, to correct the variant by using a CRISPR genome editing approach, showed after 6 months a restored MYBPC3 protein expression and normalized phenotype. Please correct the misunderstanding text at lines 284-285 and the corresponding text in the column Findings/Outcomes of Table 2.
8) Line 288: Please correct “single-stranded RNA system” with “single-guide RNA system”.
9) In Table 2 there are many errors. Please correct in the Approach column “ssRNA” with “sgRNA”. The AAV9 was used in all the cited papers (in the last paper AAV9 for the Humanized mouse model of HCM and a single plasmid encoding the editor and the sgRNA for HCM patient-derived cells). In the Findings/Outcomes column please indicate better the substantial outcomes, avoid to repeat “ Promising for…”. Please see also comment 7.
Minor comments
Line 39: Danon disease instead of Dannon disease.
In the Introduction I would suggest to delete the last lines 43-45.
Please cite Table 1 and Table 2 in the text.
Table 1: I would suggest to write “Mutations in genes encoding sarcomere proteins” instead of “Mutations in genes encoding sarcomeres in heart muscle.”
Lines 95-96: Please correct the phrase: “Those “sarcomere positive” have an identifiable pathogenic or likely pathogenic genetic variant and mutations in more.”
Lines 97-98: The presence of these mutations is found in 30% to 60% of HCM patients. Please insert an appropriate reference.
Line 99: Genetic etiology instead of genetic ethology.
Line 99: Please correct the reference 15 (see major comments)
Line 117: Please add the word “proteins” after sarcomere.
Line 122: You can delete “myosin heavy chain and myosin-binding protein C”. Just write the two acronymous MYH7 and MYBPC3, already mentioned some lines above.
Line 124: I would suggest to use “multiple mutations” instead of “different mutations”.
Line 126: Please correct “converting” with “converter”.
The concept described at the lines 123_125 is the same reported in the lines 133_135. Please avoid repetition. I would suggest to maintain the lines 133_135 and delete the lines 123_125.
Lines 168_169, 218_220, 580_586, 617_638, 718_725 and 727_737: The character size needs to be uniformed to the rest of the text.
In the paragraph “a. Myosin inhibitors (mavacamten and aficamten)” the CYP enzymes are mentioned. Please insert a little description of the role of these enzymes.
Line 225: Is it missing the word “maneuver” after Valsalva?
Line 246: I would suggest to change “Echo and CMR” with echocardiography and cardiovascular magnetic resonance.
Line 246: It is mentioned the acronymous LVM, however it was never introduced before. Please insert it accordingly.
Line 255: Is the reference number 52 the correct one?
Line 265: A typo to correct.
Lines 272-275. Please change “Moreover, AAV9 was used as a vector in a study on mice with HCM caused by a mutation in the MYL2 gen - mutation of aspartic acid to valine (D166V) in the myosin regulatory light chain (RLC) which is critical for proper cardiac contraction [58,59].” with “Moreover, AAV9 was used as a vector in a study on mice with HCM caused by a missense mutation in MYL2 gene encoding for the myosin regulatory light chain (RLC) which is critical for proper cardiac contraction [58,59]”.
Line 295. Not a single/unique mutation, but several GLA mutations were identified to caused FD. Please change “A mutation….” with “Several mutations were identified in in the alpha-galactosidase A (GLA) gene in Xq22 resulting….”.
Lines 310_312. I would suggest to use the term variants instead of codons.
Lines 327_328. The sentence here reported is a repetition of what described above at lines 314_315. Please delete the sentence at last lines of the paragraph.
Line 356: Please insert here the complete definition of ERTs. The second-generation of enzyme replacement therapy (ERTs).
Lines 378 and 381: Please change “mannose-6-phosphate receptor” with IGF-II-R.
Line 395: I would suggest to use the term “amenable mutation” instead of susceptible mutation
Line 446: Please correct the typo “GLA genes” with GLA gene.
In general, along all the manuscript pay attention to the correct use of acronymous and when cite it at the first time (LV, FD, AAV,PD, 6MWT, LOPD …). Be consistent along all the manuscript.
Lines 464_465: “The purpose of this study is to evaluate the safety and tolerability of escalating doses of ST-920 to ensure stable and effective FD”. The sentence is not clear. Please reformulate it.
Lines 499_501. I would suggest to modify the two sentences in one, as follows: “Mutations in the GAA gene, located on chromosome 17q25 [129], affect various stages of the enzyme (such as protein synthesis, post-translational modifications, lysosomal transport and maturation).”
Line 581: State in full what the FVC is.
Lines 595 and 610: Correct the typos.
Line 763: I would suggest to uniform the subtitle “a. Prospects for therapies” in accordance with the others subtitles.

From my point of view the quality of English is quite good.
Please pay attention to correct typos.
Round 2
Reviewer 2 Report
The Author's answers have been well addressed. The manuscript is now more nice to read and notably improved.